# Deep Learning based Automatic Segmentation of the Levator Ani Muscle from 3D Endovaginal Ultrasound Images

**Amad Qureshi**[*,1]                                    AQURESH@GMU.EDU

[1] *Department of Bioengineering, George Mason University, Fairfax, VA USA*

**Nada Rabbat**[*,1]                                    NMOHAMAD@GMU.EDU

**Ko-Tsung Hsu**[1]                                    KHSU5@GMU.EDU

**Zara Asif**[1]                                    ZASIF3@GMU.EDU

**Parag V. Chitnis**[1]                                    PCHITNIS@GMU.EDU

**Abbas Shobeiri**[2,1]                                    ABBAS.SHOBEIRI@INOVA.ORG

[2] *INOVA Fairfax Hospital, Fairfax, VA USA*

**Qi Wei**[1]                                    QWEI2@GMU.EDU

## Abstract

The Levator Ani Muscle (LAM) avulsion is a common side effect of vaginal childbirth and is linked to pelvic organ prolapse (POP) and other pelvic floor complications. Diagnosis and treatment of these complications require imaging and examining the pelvic floor, which is a time-consuming process subject to operator variability. We proposed using deep learning (DL) to automatically segment LAM from 3D endovaginal ultrasound images (EVUS) to improve diagnostic accuracy and efficiency. Over one thousand 2D axial images extracted from 3D EVUS data consisting of healthy subjects and patients with pelvic floor disorders were utilized for LAM segmentation. U-Net, FD-U-Net, and Attention U-Net were applied. The U-Net-based models had 0.84-0.86 mean Dice score, which demonstrated efficacy compared to literature in LAM segmentation. Our study showed the feasibility of using U-Net and its variants for automated LAM segmentation and the potential of AI-based diagnostic tools for improved management of pelvic floor disorders.

**Keywords:** pelvic floor muscle, ultrasound imaging, deep learning, image segmentation

## 1. Introduction

The Levator Ani Muscle (LAM) is a funnel-shaped structure responsible for supporting the pelvic floor, along with providing functionality, such as allowing structures to pass through it (Gowda and Bordoni, 2021). LAM avulsion, a common side effect of vaginal births, occurs in up to 35-36% of women after the first birth, causing pelvic organ prolapse (POP) and other pelvic floor disorders (Nygaard et al., 2008). Diagnosis of LAM avulsion and POP involves imaging the pelvic floor, usually through Magnetic Resonance Imaging or Ultrasound (US) imaging, the latter of which is more cost-effective (Woodfield et al., 2010). The interpretation of ultrasound is a challenging task, where the diagnosis can take weeks. To overcome the issues, we propose the use of deep learning (DL) segmentation methods, to automatically segment the LAM from 3D endovaginal ultrasound data – which has yet to be performed on such images – to improve diagnostic accuracy and reduce the diagnostic turnover time for patients.

---

[*] Contributed equally

## 2. Methods

### 2.1. Dataset, Preprocessing and Preparation

The 3D UVUS images were obtained in previously conducted work approved by the Institutional Review Ethics Committee (IRB) of the INOVA Health System (Asif et al., 2023). The dataset consists of 1015 2D axial images and LAM traces of 512x512 size from both healthy subjects (n=14) as well as patients with different degrees of pelvic floor deficiency (n=13). Several pre-processing steps were performed on the images to prepare the data for DL-based segmentation as shown in Figure 1. The healthy and unhealthy image data were independently split by 85% for training and 15% for testing, which resulted in combined 862 training images and 153 test images.

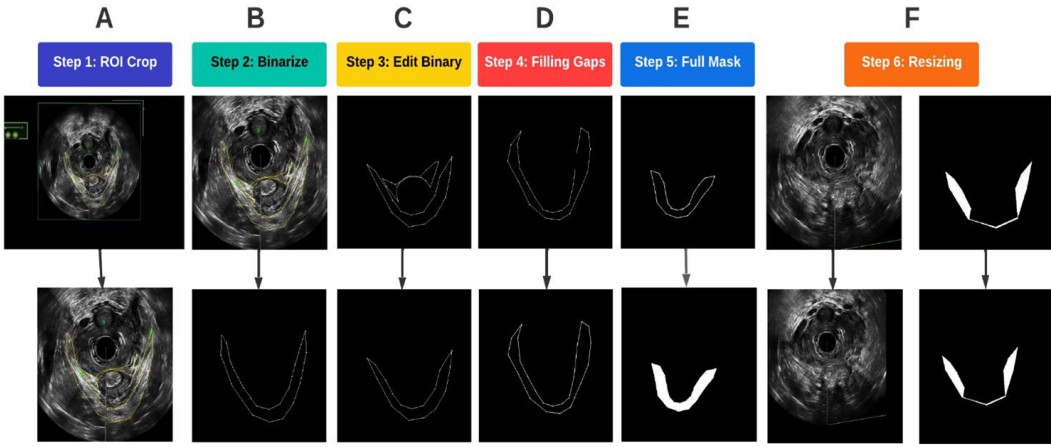

Figure 1: Flowchart of the 2D axial US image pre-processing steps

### 2.2. DL Model Configuration

Our paper explores several DL-segmentation models: U-Net, FD-U-Net, and Attention U-Net. The U-Net is a convolutional neural network (CNN) which uses a contracting path to capture context of the data and symmetric expanding path to obtain precise localization (Ronneberger et al., 2015). FD-U-Net is an extension to the U-Net in which dense connectivity into the contracting and expanding paths of network is applied (Guan et al., 2020). Attention U-Net uses attention gates to the encoding path to focus on target features of image (Oktay et al., 2018). All models were implemented with TensorFlow on a Lambda workstation with NVIDIA RTX A5000 GPU. Each model was trained in about 16min over 50 epochs with a batch size of 16. Predicted masks were subjected to Intersection over Union (IoU) and Dice accuracy metrics assessment.

## 3. Results

Our proposed study utilized U-Net, FD-U-Net, and Attention U-Net models for segmenting LAM from EVUS images. The visual results (Figure 2) show that all three models were visually similar to the ground truth (Fig. 2B). Our U-Net model (Table 1) achieved a Dice

score of 0.86 on the test data, with FD-U-Net and Attention U-Net achieving generally similar results. Furthermore, when compared to other published studies on LAM segmentation, the U-Net variants that our study implemented, especially the standard U-Net had promising results. Although comparisons were based on different datasets, such comparative analysis is important, as the proposed procedure is one of the few that dealt with LAM segmentation, and one of the only, to our knowledge, performed on EVUS.

Table 1: Comparison of DL-based LAM segmentation

| Study | Imaging Modality | Segmentation Region | Number of Images | Segmentation Method(s) | Mean Dice | Mean IoU |
|-------|------------------|---------------------|------------------|------------------------|-----------|----------|
| Proposed | EVUS | LAM | 1015 | U-Net | 0.86 | 0.76 |
|  |  |  |  | FD-U-Net | 0.84 | 0.74 |
|  |  |  |  | Attention U-Net | 0.85 | 0.75 |
| Noort, 2021 | TPUS | LAM | 100 | Recurrent U-Net | 0.65 | - |
| Feng, 2020 | MRI | LAM | 528 | CNN + MRFP | -0.61 | - |

(van den Noort et al., 2021) (Feng et al., 2020)

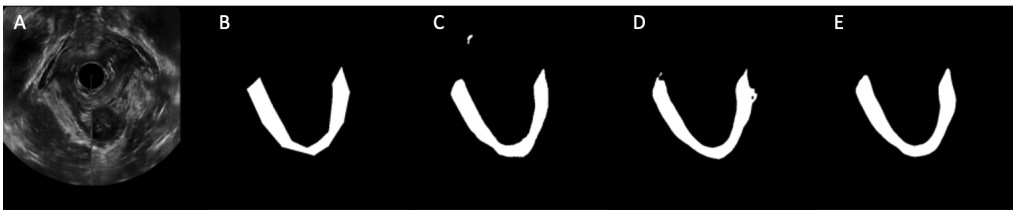

Figure 2: Example LAM segmentation results. (A) Raw EVUS image; (B) Expert traced LAM; (C) U-Net; (D) FD-U-Net; (E) Attention U-Net

## 4. Conclusion

We investigated the feasibility of using DL to segment LAM from clinical EVUS images. We found that the U-Net-based segmentation models outperform the models used in the literature to accurately segment the LAM. This study has highlighted the potential of using U-Net and its variants for the automatic segmentation of pelvic floor structures in EVUS and potentially other imaging modalities. It also has the potential of being implemented in AI-based diagnostic tools for improved management of pelvic floor disorders, especially in low socioeconomic regions, where these conditions may be underdiagnosed or misdiagnosed.

## Acknowledgments

This project is funded by the Inova-GMU Research Fund and National Institute of Health (NIH) Grant: NIH EY029715.

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
