# OpenReview forum: "Deep Learning based Automatic Segmentation of the Levator Ani Muscle from 3D Endovaginal Ultrasound Images"
_MIDL.io/2023/Short_Paper_Track — MIDL 2023 Short paper track Poster_

### Official Review · Reviewer_cRz6 · 2023-04-15
**unet-based model, maybe useful clinically**

**Rating:** 7
**Confidence:** 4

**Review:**

nice paper, clear exposition

novel application: few prior papers on this application

strong results

---

### Official Review · Reviewer_ihc8 · 2023-04-22

**Rating:** 5
**Confidence:** 5

**Review:**

The paper proposes a method to segment levator ani muscle in ultrasound images using a U-Net network. The advantage of the paper is that it utilizes and compared different backbone structures such as U-Net, FD-U-Net, and attention U-Net. However, there are several issues: (1) it is unclear whether the 3D EVUS images or the 2D axial plane images were used in the study; (2) in the experiment setup, it is unclear the training and testing data is split by subjects or not; (3) it is difficult to compare the performance of the proposed work with Noort 2021 and Feng 2020, these two works use two totally different dataset in Table 1.